# Information Technology Ambidexterity-Driven Patient Agility, Patient Service- and Market Performance: A Variance and fsQCA Approach

Rogier van de Wetering [1,*], Rachelle Bosua [1], Cornelis Boersma [2,3] and Daan Dohmen [2,4]

1 Department of Information Sciences, Open University of the Netherlands, 6419 AT Heerlen, The Netherlands; rachelle.bosua@ou.nl
2 Department of Strategic Management, Open University of the Netherlands, 6419 AT Heerlen, The Netherlands; cornelis.boersma@ou.nl (C.B.); daan.dohmen@ou.nl (D.D.)
3 Health-Ecore B.V., 3700 AA Zeist, The Netherlands
4 Luscii Healthtech B.V., 3511 HG Utrecht, The Netherlands
* Correspondence: rogier.vandewetering@ou.nl

**Abstract:** Modern hospitals are on the brink of a monumental change. They are currently exploring their options to digitally transform their clinical procedures and overall patient engagement. This work thoroughly investigates how hospital departments in the Netherlands can simultaneously leverage their strategic exploration of new IT resources and practices and exploit their current IT practices, i.e., IT ambidexterity, to drive digital transformation. Specifically, we investigate IT ambidexterity's role in shaping patient agility at the departmental level, i.e., the ability to sense patients' needs and respond accordingly. In this study, we use the dynamic capability view as our theoretical lens to develop a theoretical model with associated hypotheses and test it using cross-sectional survey data from 90 clinical hospital departments in the Netherlands. We use partial least squares (PLS) structural equation modeling (SEM) and a Fuzzy-set qualitative comparative analysis (fsQCA) approach for our analyses. This study shows that IT ambidexterity positively influences patient agility, providing a foundation for the achievement of high patient service and market performance. Furthermore, this study's outcomes show that IT ambidexterity is present in each configuration following the fsQCA analyses, showcasing the vital role of a dual strategic approach to IT practices. The study outcomes support the theorized model and the subsequently developed IT-driven patient agility framework and illuminate how to transform clinical practice and drive patient agility.

**Keywords:** IT ambidexterity; IT exploration; IT exploitation; digital transformation; patient agility; dynamic capability; hospitals; patient service performance; market performance; framework; fsQCA

## 1. Introduction

There are distinct pressures on hospitals worldwide. Think, for instance, about increased demands from patients (e.g., service, self-care tools, coordination, holistic care), but also the increased workforce pressures and competition for diagnostic technology, specialized facilities and capabilities, and the increased acuity of acute patient populations [1]. Given these pressures, strategic organization, planning, and effective execution are vital, and clinical and administrative health information technology (HIT) investments are of instrumental importance. Therefore, HIT plays a crucial role in the daily medical practice of hospitals [2,3].

Technologies such as the electronic medical record (EMR), decision-support systems, big data analytics, artificial intelligence, ubiquitously available health data, the Internet of Things, and social media apps are only a handful of the innovative technologies that are currently changing and shaping hospitals' healthcare practices [4–7]. Moreover, a recent

report on the Dutch healthcare system also highlights the crucial role of HIT in addressing the capacity planning challenges and labor market issues [8].

Using the strengths and possibilities of these disruptive forces, modern hospitals can shape their digital transformation to enhance patient service innovation and engagement [9].

Furthermore, many hospitals are embracing the 'digital transformation' process as they explore the most suitable digital options to transform their clinical procedures and overall patient engagement [4,10,11].

In addition, hospitals embrace patient-centeredness while anticipating turbulent market conditions and regulatory pressures. In this process, hospitals leverage innovative HIT to enhance efficiencies, deliver high-quality care by effectively deploying their HIT assets, resources, and organizational capabilities, and focus on state-of-the-art patient service delivery [12–14]. However limited attention has been paid to HIT's role in developing specific organizational capabilities to adequately respond to patients' needs and wishes, i.e., patient agility [15–17] and the process of leveraging HIT to enhance patient satisfaction and services and drive the departments' performance [18–20]. Thus, substantial gaps remain in the extant literature. Therefore, understanding the facets that drive HIT investment's benefits in clinical practice is valuable.

*Current Literature Limitations and Research Question*

The current paper, therefore, addresses two critical limitations. First, this paper unfolds how hospital departments—which are responsible for patient care delivery—can develop the ability to balance 'exploration' and 'exploitation' in IT resource management, i.e., IT ambidexterity [21], to drive a hospitals' patient agility, conceptualized as a dynamic capability. Hence, IT ambidexterity is crucial in addressing patients' and employees' needs. Gaining these insights is important, as there seems to be less consensus about the pivotal role of IT resources in developing these dynamic capabilities, which offer organizations the ability to deliver distinctive and mobile business services, and to anticipate market disruptions and business changes [22–24]. Second, this study shows the impact of patient agility on the department's patient service and market performance.

Focusing on ambidexterity and agility benefits at the department levels is crucial; this has seldom been explored [21,25–27], as the literature is predominately focused on the organizational level. This study, therefore, foresees that IT ambidexterity will enable the hospital departments to adequately 'sense' and actively 'respond' to patients' needs, demands, and opportunities within a turbulent and fast-paced hospital ecosystem context [23,24,28]. Furthermore, gaining these insights is essential, as hospitals are actively exploring their digital innovation options and transforming their clinical processes and interactions with patients using digital technologies [11]. In addition, the Dutch healthcare system is governed by various healthcare-related acts, including the Health Insurance Act, which covers short-term general practitioners, hospital and mental care, and medication [29]. Furthermore, hospitals in the Dutch healthcare system are bound to turnover ceilings agreements between hospitals and health insurance companies. The Dutch Healthcare Authority (NZa), an autonomous administrative authority falling under the Dutch Ministry of Health, Welfare, and Sport, also ensures that these agreements now focus on patient-centered value-creation rather than production volumes. As such, shaping departmental patient agility in Dutch hospital practice is very relevant.

Against this background and the current gaps in the literature, this paper's main objective is to examine whether IT ambidexterity contributes to higher patient agility levels and the department's patient service performance. Hence, this research attempts to address the following research questions:

1. To what extent does IT ambidexterity affect the hospital departments' patient agility and, thus, its ability to timely and adequately sense and respond to the patient's needs and demands? Furthermore;
2. What is the role of patient agility in converting the contributions of IT ambidexterity to the department's patient service and market performance?

To address these two research questions, we built upon the dynamic capabilities view (DCV) [30–32] and the ambidexterity scholarship [33] and developed a model to conceptualize a theoretical model, including three hypotheses. These hypotheses were empirically tested using cross-sectional data from 90 hospital departments in the Netherlands. We used a multi-method approach by first analyzing the data and testing the hypotheses using a composite-based approach. We also used a complementary fuzzy-set qualitative comparative analysis (fsQCA) to gain additional insight into the conditions in which hospital departments can achieve high levels of patient sensing and responding abilities.

The paper is structured as follows. First, the theoretical background highlights this study's relevant theories and concepts. Then, the following section shows the conceptual model and hypotheses development. The methodology, data analyses, and results follow this section. Following these sections, this study presents a framework for IT-driven patient agility and concludes with a discussion.

## 2. Theoretical Context

### 2.1. The Concept of IT Ambidexterity

Ambidexterity refers to the simultaneous alignment of 'exploration' and 'exploitation' by organizations, which can provide sustained competitive benefits [33]. Within information systems research, IT ambidexterity can be conceived as " . . . the ability of firms to simultaneously explore new IT resources and practices (IT exploration) as well as exploit their current IT resources and practices (IT exploitation)", Lee et al. [21]. Hence, IT exploration concerns the organization's efforts to pursue new knowledge and IT resources [21,34]. On the other hand, IT exploitation captures the extent to which organizations exploit existing IT resources and assets, e.g., reusing existing IT applications and services for patient services and reusing existing IT skills [21,35].

IT ambidexterity is a key strategic priority and has attracted serious attention over the years. IT ambidexterity ensures that the right IT resources are available in the right place at the right time. Moreover, it allows organizations to strategically manage the portfolio of digital roles and skills to retain institutional knowledge while leveraging mutually interchangeable resources where appropriate.

In practice, the simultaneous alignment of IT resources is crucial in forming digital-driven capabilities [10,12,36], especially in healthcare [37]. Furthermore, the appropriate allocation of resources in hospital departments is crucial, so that anticipated and unanticipated needs can be met. However, the imbalance between exploration and exploitation can lead to suboptimal business results [38]. Therefore, organizations need to continuously adapt existing IT resources to the current business environment and demands and focus on developing IT resources that contribute to long-term organizational benefits [16,28,34].

### 2.2. Dynamic Capabilities View and Patient Agility

The DCV is considered by many scholars to be a leading theoretical framework and is built from a multiplicity of theoretical roots [30–32]. Dynamic capabilities can be considered "the organizational and strategic routines by which firms achieve new resource configurations as markets emerge, collide, split, evolve, and die" [39]. Within the Dynamic Capabilities Theory (DCT) framework, organizations seek to balance strategies to remain stable when delivering distinct current business services and mobile to anticipate and effectively address market disruptions and business changes [23,32]. The DCV, thus, regards the environment as a crucial element that needs to be considered when deploying the firm's strategy.

Dynamic capabilities allow firms to remain stable in the delivery of distinctive and mobile current business services to anticipate and effectively address market disruptions and business changes [23,40]. These dynamic capabilities have been defined and conceptualized as sets of measurable and identifiable routines that have been widely validated through empirical studies [32,41,42].

Organizational agility, or a 'sense-and-respond' capability, has been defined and conceptualized in many ways and through various theoretical lenses in the IS literature [43,44]. It is also conceived as a manifested type of dynamic capability [23]. Moreover, the concept is influential among agility studies published in the management and information systems journals; see, for instance, [21,43].

Organizational agility can be conceived as a dynamic capability if "they permit organizations to repurpose or reposition their resources as conditions shift" [45]. Organizational agility allows organizations to respond to changing conditions while proactively enacting a dynamic environment regarding customer demands, supply chains, new technologies, governmental regulations, and competition [23]. The extant literature has conceptualized organizational agility as a higher-order construct [21,24,28]. Two organizational routines can be synthesized from the extant literature: 'sensing' and 'responding' to business events in capturing business and market opportunities. Hence, this article perceives patient agility as a higher-order manifested type of dynamic capability that allows hospital departments to adequately 'sense' and 'respond' to patient needs, demands, and opportunities within a turbulent and fast-paced hospital ecosystem context [16,23,24,27].

### 2.3. Hypothesis Development

IT can facilitate hospitals' capability-building and gain IT business value in the current turbulent market [28,43,46,47]. However, IT business value does not result from the isolated deployment of (non)IT resources and competencies. Instead, it seems to emerge from the complementarity to assimilate and re-align the IT resource portfolio to the changing business needs and demands [2]. Therefore, hospital departments must embrace an ambidextrous IT implementation strategy so that the short-term exploitation of (existing) IT resources is balanced with an exploratory mode that drives IT-driven business transformation [48].

IT exploration is explicitly about experimenting with and using new IT resources (e.g., clinical decision-support systems, big data analytics, clinical analytics, Internet of Things, and social media) to provide a basis to reshape processes and overall patient engagement. In the context of digitizing, IT exploration can help to identify and obtain digital technologies and critical IT skills that contribute to the department's strategic ambitions and plan. In addition, IT exploration facilitates hospital departments using new digital technologies to adequately adapt and adjust to patients' changing needs and wishes [21,36]. On the other hand, IT exploitation focuses more on deliberately enhancing the current IT resources. For instance, reusing or redesigning the current EMR for new patient service development and ensuring hospital-wide accessibility to clinical patient and medical imaging data and information [10,14,36].

Furthermore, IT exploration enables departments to reuse existing modular and compatible IT-infrastructures and software components and integrate them with their daily business operations and clinical practices [35,47,49]. Thus, IT exploration offers hospital departments the ability to make deliberate decisions, enhance their sensing and responding abilities, and co-evolve with the rapidly changing healthcare market [49]. However, hospital departments are better equipped to improve agility when the seemingly opposing modes of IT exploration and exploitation and, thus, the trade-off approach [22], are in sync [33,49]. In addition, the simultaneous engagement of this seemingly opposing mode of operandi ensures a clear understanding of the business, clinical, and technology contexts, and articulates how IT resources can provide value and achieve efficient, cost-effective business operational objectives.

Based on the above, this study foresees that IT ambidexterity will enable the hospital department's ability to adequately 'sense' and 'respond' to patient needs, demands, and opportunities within a turbulent and fast-paced hospital ecosystem context [23,24,28], and defines the following:

**Hypothesis 1 (H1):** *IT ambidexterity will positively enhance the patient agility of the hospital department.*

Hospitals need to deal with many strategic, organizational, and social challenges, and it is well understood that focusing on increasing patient service performance is crucial to obtaining competitive value and realizing the hospitals' ambitions and strategies [4,50]. Hospital departments can create service value for their patients by leveraging their ability to use their key resources and organizational abilities [51]. It is essential to comprehend patients' needs, preferences and wishes to provide patients with compelling healthcare propositions and services [52]. This line of reasoning is also advocated by [53,54]. Hospital department managers and decision-makers can better adopt the patient value perspective that directs the subsequent resource and sensing and responding abilities, i.e., patient agility deployment, to achieve high service performance levels. Hospital departments can achieve high patient service performance and value in the turbulent healthcare environment [55].

Effective IT-driven patient agility provides innovative ways for clinicians to improve clinical communication, remotely monitor patients, and improve clinical decision support [14,56], improving the patient treatment process and, ultimately, the quality of medical services [56,57]. As a result, hospital departments with strong patient agility are more likely to provide service flexibility, high-quality and timely services, achieve patient satisfaction, and improve the accessibility of medical services [24,36,58]. Thus, patient agility enables departments to enhance their patient service performance and ultimately strengthen their market performance [59–61]. Hence, this study hypothesizes that:

**Hypothesis 2 (H2):** *Patient agility will positively enhance the hospital department's patient service performance.*

Turbulent conditions have profound implications for hospital departments and demand new responses. The current macroeconomic trends highlight that it is imperative for hospitals to be strategically resilient and focus on digital innovation to drive market performance. Hence, hospitals with a better patient service performance will be better equipped to build a positive branch image, differentiate their patient services as 'personalized' healthcare propositions, and attain the desired business growth and market share. [46,51]. In this process, the engagement of stakeholders across all departmental levels is crucial. The pursuit of IT exploration and IT exploitation creates an organizational cascade effect, driving patient agility. Patient agility is a crucial driver for patient service performance that helps to strengthen the department's market performance [59–61]. Hence, based on the above, this study defines the final hypothesis.

**Hypothesis 3 (H3):** *Patient service performance positively impacts the hospital department's market performance.*

Figure 1 summarizes the theoretical model with the associated hypotheses.

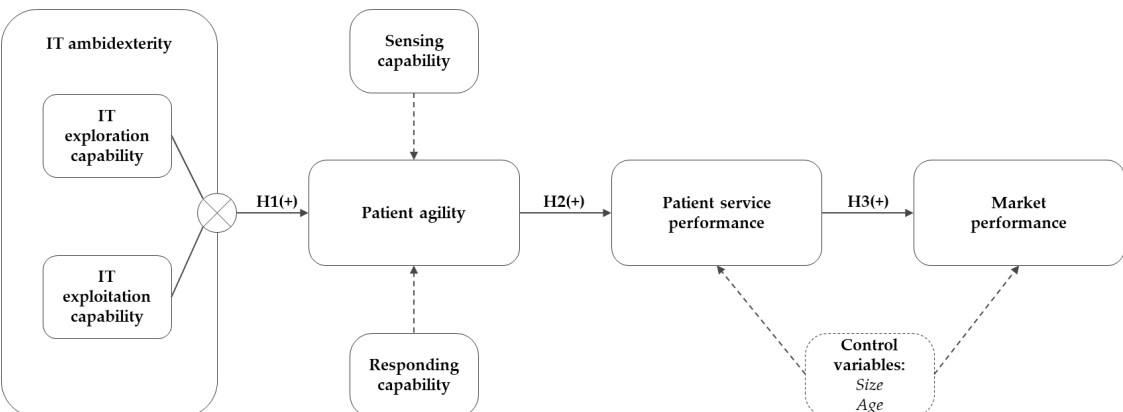

**Figure 1.** Theoretical model and hypotheses.

## 3. Methods

### 3.1. Data Collection Procedure

Survey data were systematically collected using an online survey that contained all questions to test the study's model and hypothesized relationships (c.f., Figure 1). The survey was pretested on multiple occasions by five Master's students, six medical practitioners, and scholars to improve the survey items' content and face validity. All medical practitioners had sufficient knowledge and experience to effectively assess the survey items and provide valuable improvement suggestions. The data were finally cross-sectionally collected during a field study. The target population were (clinical) department heads and managers, team leaders, and doctors under the assumption that, at the hospital department level, these respondents were actively involved in contact with patients, or at least had intelligible insight into the department's patient interactions and its use of IT. Data were conveniently sampled from Dutch hospitals through the Master's students' professional networks within Dutch hospitals. The data were collected between 10 November 2019 and 5 January 2020. Anonymity was guaranteed for the respondents. This study used 90 complete survey responses for the final analyses. Within the obtained sample, 28.9% of the respondents worked in a university medical center, 41.1% in a specialized top clinical (training) hospital, and 30% worked in general hospitals; see Table 1 for the demographics of the participating hospital departments. There are approximately 100 hospitals in the Netherlands. Eight are academic hospitals/university medical centers, twenty-six are top clinical training hospitals, and the majority are general hospitals.

Most survey respondents were medical heads/chairs of the department (51.1%), 24.4% were practicing doctors, 11.1% were department managers, and 13.3% held other positions, such as specialized oncology nurses. Table 2 provides an overview of the survey responses per medical department. As can be gleaned from the Table, a wide variety of departments participated.

Therefore, this research targeted single respondents and was sensitive to possible biases. Furthermore, possible method bias was accounted for, following specific guidelines by Podsakoff et al. [62]. Hence, this study accounted for possible non-response bias using a *t*-test (between early and late respondents) to assess whether there was a significant difference in the Likert scale questions' responses. No significant difference could be detected. Finally, Harman's single-factor analysis was applied using exploratory factor analysis (in using IBM SPSS Statistics™ v 28.0.1.0) to restrain possible common method bias [62]. Hence, the current study sample was not affected by method biases, as no single factor attributed to most of the variance.

**Table 1.** Demographics of participating hospital departments.

| Element | Category | Frequency | Percentage |
|---|---|---|---|
| Hospital type | University medical center | 26 | 28.9% |
| | Top clinical training hospital | 37 | 41.1% |
| | General Hospital | 27 | 30% |
| Department age | 0–5 years | 23 | 21.5% |
| | 6–10 years | 17 | 15.9% |
| | 11–20 years | 20 | 18.7% |
| | 21–25 years | 6 | 5.6% |
| | Over 25 years | 24 | 22.4% |
| Number of patients | <4000 | 25 | 23.4% |
| | 4000–6500 | 21 | 19.6% |
| | 6500–9000 | 12 | 11.2% |
| | 9000–11500 | 12 | 11.2% |
| | 11,500–14,000 | 11 | 10.3% |
| | >14,000 | 26 | 24.3% |

**Table 2.** Survey response per medical department.

| Department | # Responses | % of Total |
| --- | --- | --- |
| General Internal Medicine | 2 | 2% |
| Anesthesiology | 3 | 3% |
| Pharmacy | 1 | 1% |
| Cardiology | 7 | 8% |
| Cardiothoracic surgery | 1 | 1% |
| Surgery | 6 | 7% |
| Dermatology | 3 | 3% |
| Geriatrics | 1 | 1% |
| Infectious diseases | 1 | 1% |
| Intensive Care Adults | 5 | 6% |
| Pediatrics | 8 | 9% |
| Neonatology | 1 | 1% |
| Clinical immunology and rheumatology | 2 | 2% |
| Clinical Oncology | 2 | 2% |
| Lung diseases | 2 | 2% |
| Gastrointestinal and liver diseases | 2 | 2% |
| Neurosurgery | 2 | 2% |
| Neurology | 3 | 3% |
| Kidney diseases | 3 | 3% |
| Ophthalmology | 2 | 2% |
| Orthopedics | 5 | 6% |
| Psychiatry | 2 | 2% |
| Revalidation | 1 | 1% |
| First aid | 4 | 4% |
| Urology | 1 | 1% |
| Vascular medicine | 2 | 2% |
| Obstetrics/Gynecology | 8 | 9% |
| Medical imaging | 5 | 6% |
| Day treatment | 3 | 3% |
| Radiotherapy | 1 | 1% |
| Paramedic | 1 | 1% |
| Total | 90 | 100% |

*3.2. A Composite-Based Approach Using Partial Least Squares SEM*

This study relied on a composite-based approach to structural equation modeling (SEM), approximating composites rather than factors. The research model was assessed using a Partial Least Squares (PLS) SEM application, i.e., SmartPLS v 3.2.9. [63] for both the reliability and validity of the constructs (i.e., 'measurement model') and the hypothesized relationships' assessment as part of the 'structural model' evaluation [64]. PLS is a commonly preferred method when research models include mediation and when the study's nature is exploratory and emphasizes prediction-oriented work [65]. PLS is the best option for assessing models when both reflective and formative constructs (first or second-order) are involved [66], as was the case in our model.

Following Hair et al. [65], we checked the current sample's suitability using G*Power for power analyses [67], although PLS is typically recommended for relatively small sample sizes. As input parameters, this study assumed a standard 80% statistical power and a 5% probability of error, while the maximum number of predictors in the research model was two. Based on these parameters, a sample of at least 34 cases was needed (a priori) to detect an explained variance (i.e., $R^2$) of at least 0.21. Hence, the obtained sample is sufficient to assess the study's research mode and obtain stable PLS outcomes.

### 3.3. Measures, Items, and Composite Operationalization

This study tried to use empirically validated measures where possible. This study also only includes suitable measures for departmental-level analyses. IT ambidexterity is operationalized using the item-level interaction terms of IT exploration (ITEXPLORE) and IT exploitation (ITEXPLOIT) [21,33]. Items were adopted from [21]. Both constructs were reflectively modeled as latent constructs [68].

Patient agility was conceptualized as a higher-order dynamic capability comprising the dimensions 'patient sensing capability' and 'patient responding capability' [24,28,43]. This study adopted five measures for these two capabilities based on Roberts and Grover [24]. In addition, patient agility was modeled as a reflective-formative type II model [69]. This is an emergent (formative) construct with underlying first-order reflective constructs.

This study builds on the concept of IT-business value creation [51,70–72] to conceptualize patient service performance (PSP). Thus, consistent with the balanced evaluation perspectives, patient service performance is represented by three measures, i.e., enhanced quality, improved accessibility of medical services, and achievement of patient satisfaction [46,51,73]. The construct was modeled reflectively; thus, as a latent construct. Finally, market performance was measured reflectively using four items, i.e., retaining existing patients, attracting new patients, building a positive branch image, and attaining desired market share [46,55,74].

A commonly used classification was used for the Likert scale, as no archival data existed to quantify competencies and capabilities under investigation [75]. Hence, we used a scale ranging from 1 (strongly disagree) to 7 (strongly agree) for each item. In addition, following prior IS and management studies, we controlled for 'size' (full-time employees), operationalizing this measure using the natural log (i.e., log-normally distributed) and 'age' of the department (5-point Likert scale 1: 0–5 years; 5: 25+ years). Table 3 includes all the constructs' items.

**Table 3.** Constructs, items, and reliability statistics.

| Construct | | Measurement Item | λ | μ | Std. | Reliability Statistics |
|---|---|---|---|---|---|---|
| ITEXPLORE | | Please indicate the ability of your department to: (1. Strongly disagree–7. Strongly agree) | | | | |
| | EXLR1 | Acquire new IT resources (e.g., potential IT applications, critical IT skills) | 0.86 | 4.01 | 1.67 | CA: 0.79 CR:0.86 AVE:0.60 |
| | EXLR2 | Experiment with new IT resources | 0.92 | 3.81 | 1.62 | |
| | EXLR3 | Experiment with new IT management practices | 0.89 | 3.43 | 1.62 | |
| ITEXPLOIT | EXPT1 | Reuse existing IT components, such as hardware and network resources | 0.91 | 5.29 | 1.28 | CA:0.85 CR:0.90 AVE:0.68 |
| | EXPT2 | Reuse existing IT applications and services | 0.94 | 5.18 | 1.32 | |
| | EXPT3 | Reuse existing IT skills | 0.95 | 5.13 | 1.25 | |

**Table 3.** *Cont.*

| Construct | | Measurement Item | λ | μ | Std. | Reliability Statistics |
|---|---|---|---|---|---|---|
| | | Indicate the degree to which you agree or disagree with the following statements about whether the department can: (1–strongly disagree 7–strongly agree) | | | | |
| Sensing | S1 | We continuously discover additional needs of our patients of which they are unaware | 0.89 | 4.10 | 1.66 | CA:0.89 CR:0.92 AVE:0.71 |
| | S2 | We extrapolate key trends for insights on what patients will need in the future | 0.77 | 4.43 | 1.63 | |
| | S3 | We continuously anticipate our patients' needs even before they are aware of them | 0.89 | 4.03 | 1.68 | |
| | S4 | We attempt to develop new ways of looking at patients and their needs | 0.79 | 4.72 | 1.52 | |
| | S5 | We sense our patient's needs even before they are aware of them | 0.86 | 3.94 | 1.66 | |
| Responding | R1 | We respond rapidly if something important happens with regard to our patients | 0.93 | 4.52 | 1.50 | CA:0.91 CR:0.93 AVE:0.89 |
| | R2 | We quickly implement our planned activities with regard to patients | 0.91 | 4.52 | 1.42 | |
| | R3 | We quickly react to fundamental changes with regard to our patients | 0.92 | 4.54 | 1.53 | |
| | R4 | When we identify a new patient need, we are quick to respond to it | 0.87 | 4.11 | 1.62 | |
| | R5 | We are fast to respond to changes in our patient's health service needs | 0.87 | 4.76 | 1.71 | |
| | | We perform much better during the last 2 or 3 years than comparable departments from other hospitals in: (1. Strongly disagree–7. Strongly agree). | | | | |
| PSP | PSV1 | Achieving patient satisfaction | 0.83 | 4.98 | 1.32 | CA:0.75 CR:0.85 AVE:0.66 |
| | PSV2 | Providing high-quality service | 0.85 | 5.28 | 1.25 | |
| | PSV3 | Improving the accessibility of medical services | 0.75 | 4.80 | 1.33 | |
| Market performance | | We perform much better during the last 2 or 3 years than comparable departments from other hospitals in: (1. Strongly disagree–7. Strongly agree). | | | | |
| | MP1 | Retaining existing patients | 0.76 | 5.25 | 1.36 | CA:0.80 CR:0.86 AVE:0.66 |
| | MP2 | Attracting new patients | 0.76 | 4.94 | 1.39 | |
| | MP3 | Building a positive branch image | 0.83 | 5.37 | 1.28 | |
| | MP4 | Attaining desired market share | 0.78 | 4.87 | 1.38 | |

## 4. Results

### 4.1. Measurement Model Analyses Using PLS

Three tests were carried out to assess the SEM model's measurement model through SmartPLS v 3.3.3. [63], i.e., (1) internal consistency reliability, (2) convergent validity, and finally (3) discriminant validity test [64,65].

Cronbach's alpha and the composite reliability estimation show that all values are above the 0.7 threshold, demonstrating sufficient reliability [64]. This study also assessed construct-to-item loadings. None of the items had to be removed as all loadings were above 0.70 [76]. The average variance extracted (AVE) values were used to assess convergent validity. The threshold for acceptable values was 0.50 [77]. All the obtained AVE values from SmartPLS exceeded this threshold.

Finally, discriminant validity was assessed through three well-known but different tests. In the first step, cross-loadings were investigated. A high cross-loading, i.e., correlations of items (related to one specific latent construct) on other constructs, can negatively

impact discriminant validity [78]. Outcomes show that all items loaded substantially more strongly on their intended constructs. Assessment of the Fornell–Larcker criterion was used as a second step. In this process, the square root of the AVEs of all constructs is compared with cross-correlation. This analysis shows that the square root values are higher than the correlation with other latent constructs [65]; see the diagonal values in Table 4.

**Table 4.** Discriminant validity assessment.

| | Assessment of the Fornell-Larcker Criterion | | | | | | Assessment of HTMT | | | | | |
|---|---|---|---|---|---|---|---|---|---|---|---|---|
| | 1 | 2 | 3 | 4 | 5 | 6 | 1 | 2 | 3 | 4 | 5 | 6 |
| 1. EXPLO | 0.94 | | | | | | 1. EXPLO | | | | | |
| 2. EXPLR | 0.48 | 0.89 | | | | | 2. EXPLR | 0.54 | | | | |
| 3. PSC | 0.37 | 0.51 | 0.84 | | | | 3. PSC | 0.37 | 0.56 | | | |
| 4. PRC | 0.30 | 0.33 | 0.52 | 0.90 | | | 4. PRC | 0.30 | 0.36 | 0.57 | | |
| 5. PSP | 0.28 | 0.29 | 0.35 | 0.47 | 0.81 | | 5. PSP | 0.34 | 0.36 | 0.42 | 0.56 | |
| 6. MP | 0.24 | 0.17 | 0.11 | 0.12 | 0.57 | 0.85 | 6. MP | 0.29 | 0.22 | 0.17 | 0.14 | |

Note: EXPLR: IT exploration; EXPLO: IT exploitation; PSC: patient sensing capability; PRC: patient responding capability; PSP: patient service performance; MP: market performance.

Additional evidence for discriminant validity was found in a final step by subjecting the data to heterotrait–monotrait (HTMT) metric analysis [79]. The results show acceptable HTMT outcomes far below a conservative 0.90 upper bound. The higher-order (formative) construct of patient agility was assessed using variance inflation factors (VIFs) values for the constructs of patient-sensing and patient-responding capability. These VIF-values were well below the conservative threshold of 3.5. Hence, no multicollinearity was present within the research model [80].

### 4.2. Hypotheses Testing

This study used a non-parametric bootstrap resampling procedure using 5000 iterations [64] to test the hypotheses. Hence, support was found for the first hypothesis, i.e., IT ambidexterity positively impacts patient agility ($\beta = 0.48$; $t = 6.48$; $p < 0.0001$). Hence, the results reflect that the simultaneous strategic pursuit of exploration and exploitation of IT resources is a key driver of the department's ability to develop new digital-enabled processes, digitally transform clinical practice, and anticipate patients' needs.

The final hypotheses, H2 and H3, can also be accepted, as patient agility positively influences patient service performance ($\beta = 0.46$; $p < 0.0001$), and service performance is a crucial antecedent of market performance ($\beta = 0.54$; $p < 0.0001$). Thus, patient agility contributes to the achievement of high-quality and timely services, achieving patient satisfaction, and improving the accessibility of medical services to attain the desired growth levels. These results are consistent with the empirical and conceptual work of [27,28,43,44,46].

We followed specific guidelines to investigate the model's imposed mediation effects [51]. Outcomes show that patient agility 'fully' mediates the effect of IT ambidexterity on patient service performance [64,81]. In addition, patient service performance fully mediates the effect of patient agility on market performance. The control variables show non-significant effects.

The explained variance in patient agility is 23% ($R^2 = 0.23$), 22% of the variance in patient service performance ($R^2 = 0.22$) and 29% for market performance ($R^2 = 0.29$). A subsequent blindfolding assessment for the endogenous latent constructs using Stone-Geisser values ($Q^2$) showed that the model has predictive power [64]. The $Q^2$ values far exceed 0, i.e., patient agility ($Q^2 = 0.21$), patient service performance ($Q^2 = 0.14$) and ($Q^2 = 0.15$) for market performance.

Table 5 summarizes the final structural model results and outlines the estimated effect, the bias-corrected confidence intervals (Low., 2.5%–Up., 97.5%), *p*-values, and the *t*-statistic (two-tailed) of the analyses using PLS-SEM.

**Table 5.** Structural model outcomes.

| Model Path | Path Effect | Confidence Interval | *p*-Value | *t*-Value | Outcome |
|---|---|---|---|---|---|
| ITA→PA | 0.48 | CI (0.65–0.77) | <0.001 | 6.48 | H1 Supported |
| PA→PSP | 0.47 | CI (0.12–0.44) | <0.001 | 6.11 | H2 Supported |
| PSP→MP | 0.54 | CI (0.38–0.72) | <0.001 | 7.25 | H3 Supported |
| Mediation analyses | | | | | |
| ITA→PSP | 0.16 | CI (−0.07–0.37) | 0.15 | 1.45 | Insignificant |
| ITA→PSP (via PA) | 0.19 | CI (0.09–0.30) | <0.001 | 3.40 | Full mediation |
| PA→MP | −1.03 | CI (−0.07–0.37) | 0.32 | 0.99 | Insignificant |
| PA→MP (via PSP) | 0.27 | CI (0.15–0.40) | <0.001 | 4.32 | Full mediation |

Note: IT ambidexterity (ITA); patient agility (PA); patient service performance (PSP), market performance (MP).

*4.3. Configuration Analyses Using fsQCA*

In addition to variance-based analyses, we used a complementary fsQCA approach to gain additional insight into how hospital departments can achieve high levels of patient service performance by developing their sensing and responding abilities and their IT abilities [82,83]. In addition the inclusion of IT ambidexterity, we included three other relevant aspects in the analyses, as they trigger, drive and condition patient agility in clinical practice, i.e., process complexity, process intensity [84], and environmental turbulence [74].

Process complexity reflects the difficulties, uncertainties, and interdependency and, thus, the "complexity" within clinical processes [84]. Process intensity refers to the amount of patient data and information required to manage clinical processes [46,84]. Finally, environmental turbulence concerns the uncertainty concerning changes in demand (e.g., needs, wishes), competitiveness for growth and expansion in the market, and the frequency of technological disruptions and breakthroughs [74].

FsQCA is a configurational and set-theoretic approach that can unfold the present causal configurations of elements that collectively lead to outcomes of interest [32,85,86]. As such, fsQCA complements the previously outlined variance-based study results by unfolding the specific asymmetric relationship and configuration patterns between various (antecedent) constructs and specific outcomes, thereby providing a rich understanding of the data [31,87–89].

Furthermore, FsQCA allows for the predictor and outcome variables to be on a fuzzy scale. A distinction between core, peripheral, and "do not care" aspects can be made [86]. As part of the analysis, the first step defines the outcome and independent measures and accordingly calibrates them into fuzzy sets (ranging from 0 to 1, with 0 indicating the absence of set membership, while 1 denotes full membership). The independent measures are IT ambidexterity, process complexity and intensity, and turbulence. The focal outcomes are patient sensing and responding capabilities. Figure 2 depicts a Venn diagram that summarizes our current approach.

fsQCA 3.0 software [90] was used to calibrate the data and determine membership based on three particular anchors of membership using a seven-point Likert scale [91]. In addition, this study followed the calibration guidelines for generating fuzzy set-membership measures [92,93]. Hence, we generated fuzzy set-membership measures using the 75th percentile values as cut-off values for full membership, the 50th percentile as the crossover point, and the 25th percentile as the anchor value for full non-membership.

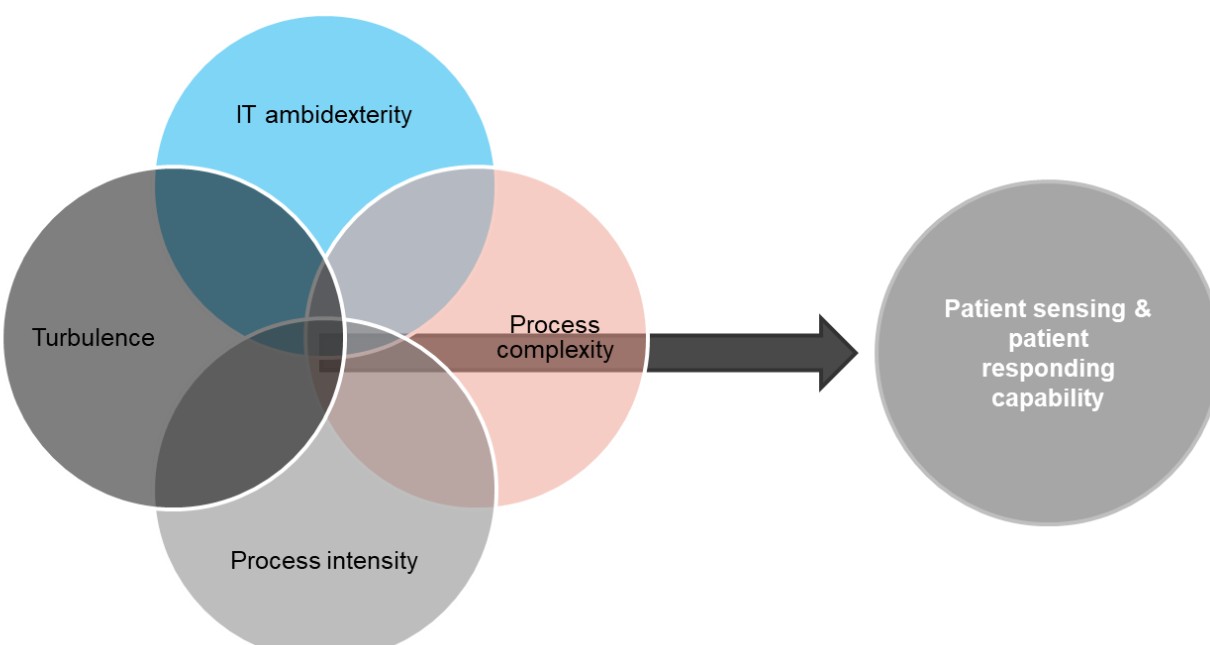

**Figure 2.** Venn diagram unfolding the theoretical constructs for the fsQCA analyses.

After the calibration process, the fsQCA software runs an algorithm to produce a truth table [91]. This Table includes all possible combinations of elements, and a row corresponds to a single combination. The number column highlights the frequency of cases of each combination. As the current sample is relatively small, a minimum frequency can be set as '1' [90,94]. However, we used combinations of elements with at least two empirical instances.

The degree of consistency is set to a threshold of 0.85, beyond the minimum value of 0.75 [86,90]. Consistency can be regarded as the degree to which a particular configuration leads to an outcome [86]. Some argue that consistency is analogous to statistical correlation with a value between 0 and 1 [95]. Another value that is crucial in fsQCA analyses is coverage. This measure shows the adequacy of a consistent subset within the solution space that estimates the focal outcome covered by the solutions [90]. Table 5 also shows solution coverage. This measure defines how the obtained fsQCA solution covers an outcome. Hence, this measure resonates with the statistical coefficient of determination ($R^2$) [95]. Table 5 also shows two coverage measures, i.e., raw coverage, which shows the proportion of outcomes of interest covered by a particular solution configuration [90], and unique coverage, which unfolds the weight of the particular solutions and the unique solution coverage [86].

Table 6 shows the outcomes of the fsQCA analysis for high levels of patient sensing and responding capability. All depicted entries concern core elements. The black circles (●) depicted in the Table demonstrate the presence of a particular condition and, in this work, these are all core elements. The crossed-out circles (⊗), on the other hand, demonstrate the absence of a particular element in the solutions space [82,90].

Table 5 shows the various unique combinations of elements that constituted high patient sensing and responding capability levels. Specifically, the results demonstrate that there were, in total, six solutions.

The first three (I, II, and III) solutions are related to high levels of patient-sensing capability, whereas the final three (IV, V, VI) show solutions related to responding capability. As can be gleaned from Table 5, IT ambidexterity is always present in each of the six solutions, supporting the core argument of our study that the dual strategic emphasis on IT exploration and exploitation is crucial in obtaining business value and fostering adaptive capabilities, i.e., patient agility. In addition, the fsQCA outcomes also show

various combinations of sufficient conditions that explain patient agility's underlying capabilities. For instance, solution I applies to hospital departments that operate under conditions with a strong IT ambidexterity (and, thus, well-developed IT exploration and exploration capability) and high process intensity. Thus, these departments have clinical processes and disciplines that require high levels of patient data and, therefore, a high degree of information-processing capacity.

**Table 6.** Configurations for the achievement of patient sensing and responding capabilities.

| Configurational Items | Solutions for Patient Sensing Capability | | | Solutions for Patient Responding Capability | | |
|---|---|---|---|---|---|---|
| | I | II | III | IV | V | VI |
| IT ambidexterity | ● | ● | ● | ● | ● | ● |
| Process complexity | | ⊗ | ● | | ● | |
| Process intensity | ● | | | | | ● |
| Turbulence | | ⊗ | ● | ⊗ | | |
| Assessment scores | | | | | | |
| Raw coverage | 0.472 | 0.227 | 0.335 | 0.315 | 0.411 | 0.421 |
| Unique coverage | 0.157 | 0.091 | 0.066 | 0.0836 | 0.068 | 0.068 |
| Consistency | 0.683 | 0.702 | 0.793 | 0.619 | 0.669 | 0.669 |
| Overall solution consistency | 0.689 | | | 0.662 | | |
| Overall solution coverage | 0.649 | | | 0.602 | | |

Contrary, solution II indicates that conditions with strong IT ambidexterity and the absence of process complexity and environmental turbulence are sufficient to predict high levels of patient-sensing ability. Thus, these departments operate under more stable conditions and work on routine, standard clinical diagnostics and care, independent of any volatility in the environment. On the other hand, solution III applies to hospital departments that work under the core conditions of strong clinical process complexity and high environmental turbulence, as both these conditions are present in the obtained solution space. Therefore, it is crucial to invest in core, and innovative IT capabilities that support exceptional data processing and IT support and flexibility to adjust and renew the infrastructure to address the required and even (environmentally and regulatory) imposed changes.

Solution IV refers to departments that, like those in solution II, work independently of any environmental turbulence. Solution V and VI apply to hospital departments that have established a strong IT ambidexterity, as this is also a core condition for the achievement of high levels of patient-responding capability. However, in solution V, we see the presence of a strong clinical process complexity and, thus, high uncertainties and interdependency within clinical processes. Under these conditions, hospitals require IT capabilities and IT interactions that support collaboration and communication and drive daily business operations and clinical practices. Solution VI, like solution I, achieves a high responding capability under the core condition of high process intensity. The simultaneous engagement of IT exploration and IT exploitation is crucial for these departments to provide value and achieve efficient, cost-effective business operational objectives.

## 5. A Framework for IT-Driven Patient Agility and Digital Transformation

Building upon this study's theoretical context and critical results, this section proposes a practical, IT-driven patient agility and digital transformation framework (c.f., Figure 3). This framework shows the key drivers and enhancement activities of patient agility in clinical practice, critical entry paths, and foundational opportunities that drive further investments that increase hospital performance.

In practice, this framework serves as a screening mechanism for hospital departments to identify potential opportunities at the intersection of attractive value creation opportunities and unique sources of IT resource leverage.

Achieving patient agility requires more than just a technical change. It requires a true shift within hospitals from people and process to technology, and this framework provides hospital decision- and policymakers with the required accelerators to deliver quality projects while still providing the flexibility needed to solve each unique challenge.

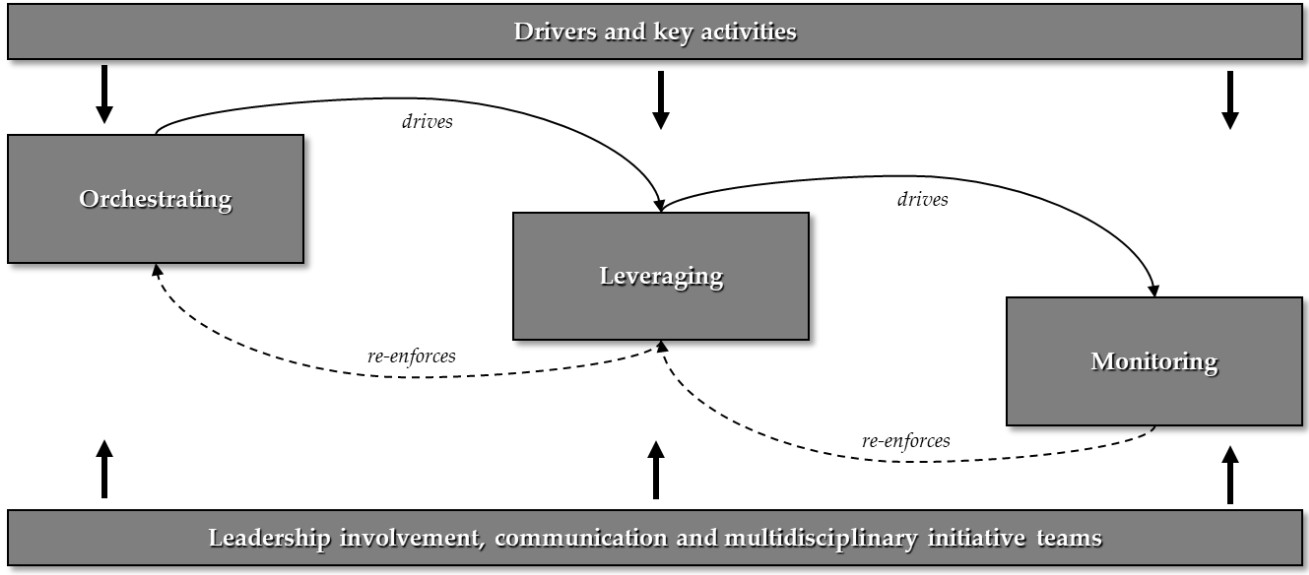

**Figure 3.** The framework of IT-driven patient agility and digital transformation in clinical practice.

This process-based framework contains three major stages, i.e., (I) orchestrating IT capabilities, (II) leveraging patient agility, and (III) monitoring service and market value, which are tightly associated with the concepts and the IT-driven value path this study investigates. At each stage, essential drivers and enhancement activities will be discussed.

*5.1. Orchestrating IT Capabilities*

At the orchestration stage, the hospital department needs to develop technical, and management skills and IT practices to align IT applications with goals and strategic priorities. Leveraging IT resources is crucial to meet new business requirements, adapt to changing market conditions, and obtain business benefits [12,55,96,97]. Additionally, departments should develop the ability to master digital technologies, drive digital transformations, and develop innovative patient-centered services and products. Hospitals also need to develop knowledge processes alongside technical skills and competencies [27]. Knowledge processes are crucial operations with hospital departments that facilitate the transformation of clinical data into patient-related insights, thereby supporting clinicians within hospitals to make informed decisions concerning diagnosis and treatment [14,57,98]. Knowledge processes enhance other organizational capabilities based on the degree of knowledge reach and richness that the organization can achieve [43,99]. In sum, orchestrating knowledge processes and digital practices is crucial to driving the productivity of hospital departments.

Key driving activities at this stage include the diagnosis of the current interplay between formal, interpersonal, and cultural mechanisms in the department to identify possible barriers to the desired strategy and change. It is also essential to engage all stakeholders, i.e., medical providers and supporting staff, to rapidly embrace the digital transformation and various (innovative) IT improvement initiatives. Likewise, the digital skills and competencies needed for success must be clearly articulated. This has important implications for the hospital department managers and decision-makers, as the individuals and teams involved in clinical practice and improvement projects need the appropriate skills and training. Therefore, the deployment of digital skills and competencies requires adequate funding to fuel this.

### 5.2. Leveraging Patient Agility

Orchestration drives the leveraging stage of the IT-driven patient agility and digital transformation framework, focusing on establishing disciplined, rigorous, and consistent sensing and responding capabilities. This study shows that developing strong patient agility capabilities is essential to providing high-quality and flexible medical services. Hence, these adaptive sensing and responding capabilities need to be cultivated through a carefully designed and executed approach to organizational change and transformation. This process requires a high engagement planning process with an extended department leadership team and a rapid cascade process for all physicians, nurses, management, and other medical providers to build understanding, ownership, accountability, and alignment.

In this stage, it is vital to develop the appropriate structures and systems to establish patient agility and facilitate its evolution over time while balancing the tensions between formal and informal structures and systems. Understanding patient agility performance by analyzing specific past experiences and the root causes of successes and failures is also crucial in this stage. Moreover, hospital departments should change the organizational and departmental culture and capabilities. Effective change campaigns simultaneously focus on changing individuals' behavior and institutional departmental features in clinical practice. The transformations that the hospital department launches should be introduced with care and in alignment with the strategic direction to be successful.

Decision-makers should reinforce rapid alignment across physicians and other medical providers, with specific commitments from all staff that become the backbone of the drive towards agility. By leveraging IT applications and knowledge processes in clinical practice, physicians can attempt to discover patients' additional needs and extrapolate key trends for insights into what patients will need in the future. This process should go hand-in-hand with the simultaneous configuration of electronic medical records or even dedicated clinical decision-support software that enables physicians to work with the system to make informed and accurate decisions concerning diagnosis and treatment. In this way, hospital operations and clinical pathways are organized to enable physicians and (specialized) nurses to react to patient needs and demands and respond to changes in the patient's health service needs. Consequently, the patient treatment process and medical quality services can be enhanced.

### 5.3. Monitoring Service and Market Value

The final stage, i.e., monitoring value, is all about measuring and acting upon what is measured. The focus should be explicitly on linking metrics upwards, downwards, and even laterally to create a monitoring system that enables learning about the interdependencies and interactions in the hospital department. The metrics should be drawn from strategic objectives and re-enforce learning about patient service and (cost-)efficiencies; incorporating this learning into the metric system is critical to continuous improvement. The monitoring system serves as a credible guide to action in clinical practice. Identifying linkages provides physicians and department managers a clear line of sight regarding how their metrics relate to other departments, even beyond the hospital's boundaries, making the implications of various choices and actions explicit.

Initially, the focus should be on a limited diversified set of the hospital department's long- and short-term goals to initiate such a system. However, going through the first stages, i.e., orchestration and leveraging, is critical and enables the transformation's success without limiting the departments' responsiveness and ability to adapt to change. Moreover, detailing the roles, tasks, and responsibilities in daily practice is essential to ensuring an aligned strategy.

This stage is essential for hospital departments that want to deploy their resources against future strategic objectives and the ambition to develop a what-if ability to look ahead. Finally, the monitoring system should be flexible, allowing for easy revision as the health landscape changes.

In this stage, but also throughout the other stages, strong communication can increase excitement about enhancing clinical processes and strengthening the quality of the services. A motivated leadership is required to drive the transitions, and multidisciplinary teams (involving physicians, nurses, and other medical providers) should drive improvement initiatives. This stage is a crucial reinforcement of the previous stages, ensuring that a feedback loop is established, and a continuous improvement is implemented within the hospital department.

It should be acknowledged that the achievement of a high patient service performance and market performance stems from the specific interplay between IT ambidexterity and patient agility within the rapidly changing healthcare ecosystem, as also demonstrated in the fsQCA configurational analyses. This means that performance enhancement depends on the various contingency elements that are considered when formulating an improvement strategy [31,32].

## 6. Discussion and Concluding Remarks

Hospitals worldwide are on the brink of a monumental change due to all the disruptive forces and pressures, especially now, during the COVID-19 pandemic. Therefore, developing adequate digital skills and investing in the necessary abilities for digital transform is crucial to leveraging innovative clinical practice technologies.

This study is built upon the core idea that strengthening IT ambidexterity, and thus the simultaneous engagement of IT exploration and exploitation capabilities, will drive hospitals' ability to adequately sense patient needs, wishes and behavior, respond accordingly, and contribute to hospital departments' overall patient services and performance levels. This study developed a theoretical model and three hypotheses and used an online survey with data from 90 clinical departments in the Netherlands to assess the model's theoretical assumptions. As a result, this study found support for all the hypothesized relationships and supports the positioning of a framework for IT-driven patient agility and digital transformation in clinical practice.

This work has several theoretical and managerial contributions that will be discussed next.

### 6.1. Theoretical Contributions

First, this study's central claim was that when hospital departments are ambidextrous in IT resource management, they are more likely to adequately sense and respond to patients' needs and wishes and achieve better patient service performances. The results of this study corroborate this claim based on the PLS analyses. Therefore, the results shed light on the current lacunas in the extant literature concerning the achievement of patient service performance and benefits through IT ambidexterity and patient agility. Previous work argued that ambidexterity and agility benefits at the department levels are crucial for digital transformation [21,25–27]. However, empirical evidence was lacking, and the literature is predominately focused on the organizational level.

Second, this study adds to the current knowledge base on how 'digitizing' supports the capability-building processes, facilitates patient agility, and contributes to the much-needed insights on obtaining IT value at the departmental level [20,21,25,26]. Third, in line with work by Lee et al. [21], we show that IT ambidexterity directly and positively affects the department's organizational ability to sense and respond. Moreover, this work extends previous work on IT ambidexterity and digital resource deployment in healthcare [16], showing that enhanced patient services and market performance can be obtained through a dual digitalization approach [10,12,16,36,37].

### 6.2. Practical Implications

This study has several implications for practice. First, this study shows that hospital departments should invest in their ability to balance both the organization's efforts to pursue new knowledge and IT resources and their ability to take advantage of existing IT resources and assets. Empirical results show that IT ambidextrous hospital departments

can better identify new innovative digital opportunities and patient services and enhance patient agility. Thus, this study unfolds the critical resources and abilities that hospital departments can leverage from a patient agility perspective.

Results suggest that hospitals that are committed to ambidextrously managing their IT resources are more proficient in promptly sensing and responding to patients' medical needs and demands. Therefore, decision-makers and hospital department managers can justify HIT investments as a source of IT's business value [21,71,100].

Second, many hospital departments have become highly practiced, and thus are stuck at the benefits and patient service outcomes they achieve at present. When the department's doctors and decision-makers want different or better patient service outcomes, it is first essential to diagnose the current interplay between IT ambidexterity and patient agility capabilities to identify key drivers of and barriers to the desired change. In practice, attempts to change a department's way of working by changing just one steering mechanism nearly always fail. It seems that the nature of the current system overwhelms any single change. Effective enhancement initiatives simultaneously focus on individual changing behaviors and institutional features.

Therefore, hospital department decision-makers should pay attention to end-users' psychological meaningfulness, stakeholder involvement, and adequate resourcing and infrastructures when implementing new digital technologies [101–103].

Finally, following the outcomes related to hypothesis 3 (i.e., patient service performance positively impacts the hospital department's market performance), we argue that some hospitals might outperform others in the healthcare ecosystem. This means that embracing digitalization and, hence, IT ambidexterity might lead to some hospitals becoming market leaders in their fulfillment of patient wishes and needs.

*6.3. Limitations and Concluding Remarks*

Several study limitations should be addressed. First, the current data were collected using a single informant strategy. Therefore, method bias could be a concern. Future work could use a matched-pair approach, where different respondents address independent (explanatory) and dependent constructs. Although sufficient for the current study purposes, the current sample was relatively small. Hence, a more extensive sample could contribute to the robustness of the results. Department-specific analyses could also be performed, to offer a richer and more comprehensive view of the subject matter. Finally, we only included IT ambidexterity as a key antecedent of patient agility. Future work could also include other organizational and IT capabilities (e.g., IT human, big data analytics, and artificial intelligence).

To conclude, this study provides critical insights into how hospital departments can further shape their patient agility and drive service and market performance. The study's contributions are substantial because the extant literature did not explain how the hospital department can leverage IT resources in shaping dynamic capabilities under turbulent conditions. Moreover, by looking at the composite-based and fsQCA analyses, we could identify specific conditions and unique combinations of elements that constituted high patient sensing and responding capability levels. The emerging patterns and pragmatic framework can support hospital decision-makers in exploring innovation options at the intersection of attractive value-creation opportunities and unique sources of IT resource leverage, and transform clinical processes and interactions with patients using digital technologies.

**Author Contributions:** Conceptualization, R.v.d.W.; methodology, R.v.d.W.; software, R.v.d.W.; validation, R.v.d.W., R.B., D.D. and C.B.; formal analysis, R.v.d.W.; investigation, R.v.d.W., R.B., D.D. and C.B; data curation, R.v.d.W.; writing—original draft preparation, R.v.d.W.; writing—review and editing, R.B., D.D. and C.B.; visualization, R.v.d.W.; supervision, R.v.d.W. All authors have read and agreed to the published version of the manuscript.

**Funding:** This research received no external funding.

**Institutional Review Board Statement:** This work applied our Faculty of Science research principles and, for this type of work, there is no need to go through our Research Ethics Committee (cETO) further, our university-wide advisory committee that operates under the Deans of the Faculties.

**Informed Consent Statement:** Not applicable.

**Data Availability Statement:** Participants of this study did not agree for their data to be shared publicly. Therefore, the data is not available.

**Acknowledgments:** We would like to thank Josja Willems, Reinier Dickhout, Rick Smulders, Yves-Sean Mahamit, and Renaldo Kalicharan for their valuable contributions to the data collection and for sharing their perspectives in numerous discussions during their thesis project. We would also like to express our gratitude to all the participating hospitals.

**Conflicts of Interest:** The authors declare no conflict of interest.

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
