# Peer review of "Information Technology Ambidexterity-Driven Patient Agility, Patient Service- and Market Performance: A Variance and fsQCA Approach"

_sustainability, doi:10.3390/su14074371_

Round 1

Reviewer 1 Report

Thanks for the interesting manuscript. Some suggestions:

  1. It will enhance the readability if the current long introduction has a sub section on literature/background context;
  2. It is unclear what is the overarching research design in the methodology section;
  3. It will be useful to highlight how the results answer the hypotheses;
  4. It will be helpful to include the significance of the study as well as possible future study in the conclusion. 

Look forward to the revised version. Thanks. 

Author Response

Many thanks for these extensive and thoughtful reviews on our paper and for your invitation to resubmit the paper with corrections. The comments have been most helpful in refining this work.

Reviewer 2 Report

Dear Authors,

The topic of the manuscript is interesting, but there are some issues that should be considered in order to improve the quality of the paper and make it publishable.

First of all, the authors made great effort to present the idea, keeping the logical flow throughout the manuscript. The manuscript really has a great potential.

As far as important and necessary major improvements, the comments are as follows:

  • It would be preferred that first paragraph in the introduction of the manuscript has some references to support the presented claims
  • It would be also interesting to have some introduction about Dutch healthcare system. What are its main characteristics, are they similar or different to other countries, and so on? Just to have a context and also to be aware about the possibilities to implement recommendations in other environments
  • The sample is clearly presented and justified. However, there is no information about the population. Therefore, we cannot be sure if these data are representative. Please elaborate
  • I was wondering if there are any major differences between the answers of different categories? Maybe some discussion on this matter could be done
  • The discussion should be more thoroughly elaborated, making a connection with previous research in the field. Furthermore, you should be more precise in explaining who and how can benefit from the results presented in the manuscript

Best regards

Author Response

(The authors gave the same response as above.)

Round 2

Reviewer 2 Report

All suggestions that were proposed are well addressed and appropriately incorporated in the manuscript. Therefore, I am more than happy to propose acceptance of this manuscript. My sincere congratulations to the authors and all the others involved in this process.

Best regards